# Long Noncoding RNAs Regulate the Inflammatory Responses of Macrophages

**DOI:** 10.3390/cells11010005

**Published:** 2021-12-21

**Authors:** Qing Zhao, Gaozong Pang, Lin Yang, Shu Chen, Ruiyao Xu, Wei Shao

**Affiliations:** School of Basic Medical Sciences, Anhui Medical University, Hefei 230032, China; zhaoqing@stu.ahmu.edu.cn (Q.Z.); 512620pang@gmail.com (G.P.); 1846010008@stu.ahmu.edu.cn (L.Y.); shuchen976@gmail.com (S.C.); xuruiyao@stu.ahmu.edu.cn (R.X.)

**Keywords:** long noncoding RNA, macrophage, inflammation, immune system, polarization

## Abstract

Long noncoding RNAs (lncRNAs) are defined as transcripts with more than 200 nucleotides that have little or no coding potential. In recent years, due to the development of next-generation sequencing (NGS), a large number of studies have revealed that lncRNAs function as key regulators to maintain immune balance and participate in diverse physiological and pathological processes in the human body. Notably, overwhelming evidence suggests that lncRNAs can regulate innate immune responses, the differentiation and development of immune cells, inflammatory autoimmune diseases, and many other immunological processes with distinct regulatory mechanisms. In this review, we summarized the emerging roles of lncRNAs in macrophage development and polarization. In addition, the potential value of lncRNAs as diagnostic biomarkers and novel therapeutic targets for the treatment of aberrant immune responses and inflammatory diseases are discussed.

## 1. Introduction

LncRNAs are a class of highly heterogeneous RNA molecules with lower conservation than protein-coding genes [1]. Increasing evidence shows that lncRNAs regulate gene expression at multiple levels, such as chromatin remodeling, transcription regulation, and posttranscriptional processing [2]. LncRNAs are located in the nucleus and can recruit chromatin modification complexes to DNA, thereby blocking the binding of transcription factors to their promoters or acting as transcription coactivators [3,4,5]. LncRNAs can also indirectly regulate gene expression by sponging microRNAs, which function as competing endogenous RNAs (ceRNAs) [6]. According to the relative position of the coding genes, lncRNAs can be divided into antisense lncRNAs, intronic lncRNAs, divergent lncRNAs, intergenic lncRNAs, promoter upstream lncRNAs, promoter-associated lncRNAs, and transcription start site-associated lncRNAs [7].

Macrophages are large phagocytes found in all tissues. Macrophages are essential for innate immunity, normal tissue development, homeostasis and repair of damaged tissues and are involved in multiple biological processes, such as phagocytosis of dying cells [8], pattern recognition of microorganisms or altered ligands [9], antigen processing and presentation [10], release of pro- and anti-inflammatory mediators [11], wound repair [12], and the homeostasis of adipose tissue [13]. In the past decade, many studies have shown that macrophages, which are immune cells that recognize the “plasticity” of the microenvironment, can be epigenetically programmed by signals from the tissue environment [14]. Macrophages can be polarized into two subsets with diverse functions by specific microenvironmental stimuli and signals. The first subset is classically activated macrophages (M1), and the second subset is alternatively activated macrophages (M2) [15]. However, many studies have also revealed that macrophages display a series of continuous functional states. M1 or M2 macrophages are just two extremes of the dynamically changing state of macrophages [14]. Among the various putative molecular and biological functions conferred on lncRNAs, lncRNAs play an important role in establishing and maintaining discrete and lasting epigenetic modifications [16]. An attractive point of view in epigenetic research is that the immune enhancement observed in trained monocytes and macrophages is driven by epigenetic modifications that regulate the expression of innate immune genes [17]. Additionally, innate immune cells can produce an immune memory called trained immunity or innate immune memory to deal with long-term exposure to microbial components so that they can remember transcriptional responses and even inherit these memories to offspring [18,19,20]. Within the last decade, work to decode gene functions has revealed the key role of lncRNAs in the epigenetic regulation of gene transcription [21,22]. This finding reminds us that lncRNA-mediated regulation is central to the establishment of trained immunity.

Usually, M1 macrophages can be induced by Th1 cytokines (such as interferon-γ and tumor necrosis factor-α) or bacterial lipopolysaccharide (LPS) [23]. This produces high levels of proinflammatory cytokines, such as TNF-α, interleukins (IL-1α, IL-1β, IL-6, IL-12, and IL-23), and cyclooxygenase-2 (COX-2), and low levels of IL-10 [24]. In contrast, M2 macrophages secrete anti-inflammatory cytokines such as IL-4, IL-13, IL-10, or TGF-β and play roles in anti-inflammatory processes and tissue repair [25]. Although great progress has been made in understanding the physiology and pathology of macrophages, due to the lack of strict phenotypic scoring criteria, the specific terms M1 and M2 are still controversial [26].

Recently, the critical roles of lncRNAs in regulating the inflammatory responses of macrophages have been discovered. Some lncRNAs induce inflammation and polarize macrophages into the M1 state. Other lncRNAs suppress inflammation and trigger macrophage polarization to the M2 state. Moreover, recent advances in the understanding of the role of lncRNAs in macrophage function in homeostasis and disease are discussed in this review along with the possible mechanisms underlying the regulatory link between lncRNAs and macrophage-related inflammation.

## 2. Proinflammatory Macrophage-Targeting lncRNAs

### 2.1. Carlr

Cardiac and apoptosis-related long noncoding RNA (Carlr) is a single-exon intergenic lncRNA that is 2574 nucleotides in length and consists of one exon, located between the *Spag6* and *Pip4k2a* genes on mouse chromosome 2. Ainara et al. demonstrated that the expression of Carlr was upregulated in macrophages stimulated with LPS [27]. Most Carlr transcripts moved from their resting location in the nucleus to the cytoplasm in response to LPS. In a study of intestinal inflammation, researchers cocultured human Caco-2 intestinal cells with activated macrophages and found that Carlr also showed cytoplasmic localization in these cells. After knocking out Carlr in macrophages and coculturing the knockout cells with intestinal cells, the proliferation of intestinal cells was reduced, and the translocation of NF-κB was diminished. Further studies using THP-1 cells revealed that using small interfering RNA (siRNA) to silence Carlr could suppress the expression of the *Il-1β* and *Ptgs2* genes activated by LPS. RNA immunoprecipitation (RIP) experiments showed that Carlr interacts with NF-κB p65 before entering the nucleus in THP-1 cells [28]. In this regard, it is interesting to note that Carlr may be a potential novel participant in the NF-κB inflammatory pathway and play an important role in signal transmission between intestinal cells and macrophages.

### 2.2. CHRF

LncRNA-CHRF (cardiac hypertrophy-related factor) is a 1841 nucleotide transcript. In silica-activated RAW264.7 cells, a substantial number of oxidants and cytokines were secreted, and the expression of CHRF was also upregulated. Further study revealed that lncRNA-CHRF acts as an endogenous “sponge” of miR-489 [29]. Overexpression of CHRF could change the inhibitory effect of miR-489 on MyD88, thereby triggering the inflammation signaling pathway, which can be viewed as a promising drug target for the treatment of silicosis [30].

### 2.3. FENDRR

The intergenic mouse lncRNA fetal-lethal noncoding developmental regulatory RNA (FENDRR) is a 2397 nucleotides transcript composed of seven exons transcribed from the transcription factor-coding gene *Foxf1* [31]. Xu et al. found that the human FENDRR gene is 3099 bp in length and consists of four exons [32]. In THP-1 cells stimulated by IFN-γ, phosphorylation of STAT1 increased and activated the transcription of FENDRR and M1 macrophage-specific markers, including IL-1β, TNF-α, and CXCL10. FENDRR expression further enhanced the phosphorylation of STAT1 [33] However, the specific mechanism by which FENDRR regulates STAT1 phosphorylation and promotes the polarization of M1 macrophages has not yet been reported.

### 2.4. FIRRE

The nuclear-retained and conserved lncRNA functional intergenic repeating RNA element (FIRRE) is 79,518 nucleotides in length and consists of seven exons, anchors the inactive X chromosome by maintaining H3K27me3 methylation. FIRRE consists of a unique 156-bp repeating sequence that interacts with heterogeneous nuclear ribonucleoprotein U (hnRNPU) [34,35]. In RAW264.7 cells, NF-κB subunit p65 could bind to the promoter region (-1720) of FIRRE, an NF-κB-dependent lncRNA, following LPS stimulation. Lu et al. found that hnRNPU and adenylate-uridylate–rich element (ARE) are important mRNA-stabilizing proteins [36]. FIRRE interacts with these proteins and enhances the mRNA stability of inflammatory genes, including VCAM-1 and IL12p40, by targeting the adenylate-uridylate–rich element (ARE)-mediated posttranscriptional mechanism. Finally, FIRRE actively regulates the expression of inflammatory factors [37]. Taken together, the above evidence sheds light on the important role of FIRRE in the posttranscriptional regulation of inflammatory genes.

### 2.5. GAS5

Mouse lncRNA-Growth Arrest Specific 5 (GAS5) is a lncRNA with a length of 15,590 nucleotides, can silence the expression of the chemokine CCL1, which is necessary for M2b macrophage survival [38,39]. In bone marrow-derived macrophages (BMDMs) stimulated by LPS and immobilized IgG (immune complex, IC), GAS5 expression is decreased by the nonsense-mediated mRNA decay (NMD) pathway to trigger BMDM polarization to the M2b state [40]. Hu et al. found that the expression of GAS5 is considerably increased in macrophages induced by hyperglycemia and inhibits the conversion of M1 macrophages to M2 macrophages, impairing wound healing in people with diabetes [41]. Another study of macrophages from children with pneumonia uncovered that GAS5 acts as the ceRNA of miR-455-5p to induce the polarization of M1 macrophages and increase the expression levels of IL-1 and IL-12 [42]. In addition, Ye et al. found that GAS5 expression was upregulated in oxLDL-treated THP-1 cells. GAS5 acts as a sponge of miR-221, aggravating local inflammatory responses, including increasing the expression of IL-6, IL-1β, TNF-α, and monocyte chemoattractant protein-1 (MCP1) and overactivating M1 macrophages [43]. The aforementioned findings indicate that the role of GAS5 in a variety of diseases is worthy of our attention.

### 2.6. HOTAIR

HOX transcript anti-sense intergenic RNA (HOTAIR) is a well-studied lncRNA that is 3669 nucleotides in length and consists of two exons. HOTAIR plays key roles in the immune response mediated by inflammatory genes and cytokines. In HOTAIR knockout cells, the expression of IL-6 and iNOS was significantly downregulated at both the mRNA and protein levels [44]. LPS treatment induces the expression of HOTAIR in RAW264.7 cells, primary macrophages and BMDMs. Expression of HOTAIR in macrophages stimulated by LPS is regulated by NF-κB. Additionally, HOTAIR is essential for promoting IκBα degradation induced by LPS, activation of NF-κB and nuclear translocation. Research on the expression of HOTAIR in macrophages/myeloid-derived suppressor cells (MDSCs) in the tumor microenvironment revealed that HOTAIR is highly expressed by tumor-associated macrophages. HOTAIR expression promotes the secretion of cytokines or chemical factors such as CCL2, which in turn induces tumor growth and metastasis [45]. Another study revealed that HOTAIR expression was downregulated in macrophages infected with virulent (H37Rv) strains of Mycobacterium tuberculosis but its expression was upregulated in avirulent (H37Ra) strains. ChIP-seq using an H3K4me3-specific antibody revealed that there was a difference in the percentage of H3K4me3 [45] in cells from people infected by the two strains, and the difference was related to SATB1 and DUSP4 expression. Furthermore, overexpression of HOTAIR induces chemokines, such as CXCL1, CXCL2, and CXCL3, which is related to the inhibition of the transcription of DUSP4 and SATB1 and an increase in H3K27me3 in the upstream region [46]. Rinn et al. observed that HOTAIR transrepresses the HOXD locus and recruits polycomb repressive complex 2 (PRC2) to the locus to establish the inhibitory chromatin marker H3K27me3 [3]. Tsai et al. later demonstrated that HOTAIR not only directly interacts with PRC2 at the 5′-end but also has the H3K4-targeting demethylase LSD1 [47]. These findings broaden the potential of lncRNAs as an overall epigenetic regulatory mechanism.

### 2.7. H19

LncRNA H19 is controlled by patrilineal and matrilineal imprinting [48], which is 2615 nucleotides in length and has five exons. Previous studies of cholangiocarcinoma [49] and abdominal aortic aneurysm [50] have revealed that H19 is abundant in infiltrating macrophages. H19 expression significantly upregulated the levels of the proinflammatory genes IL-6 and MCP-1. Han Yong’s studies of patients with atherosclerosis and (oxygenized low-density lipoprotein) oxLDL-stimulated RAW264.7 cells revealed that the levels of the anti-inflammatory factors IL-4 and IL-10 were increased, and the levels of the pro-inflammatory factors TNF-α and IL-1β were decreased when H19 expression was silenced. Simultaneously, the inflammatory symptoms were considerably alleviated. Further mechanistic studies revealed that H19 can upregulate the expression of miR-130b in oxLDL-treated macrophages, thus regulating inflammation, lipid metabolism, and foam cell formation [51]. Li et al. also found that in BMDMs isolated from wild-type mice, H19 exosomes from bile duct cells could promote the differentiation and activation of proinflammatory M1 macrophages and play a pivotal role in upregulating the expression of IL-6, IL-1β, COX-2, and CCl-5 [52]. Thus, studies resulting from the explosion of interest in H19 as a regulator of inflammatory genes have revealed that it is a potentially effective target for the treatment of inflammation.

### 2.8. Maclpil

Macrophage contained lymphocyte cytosolic protein 1 (LCP1)-related proinflammatory lncRNA (Maclpil), also named Gm15628, which is 643 nucleotides in length and consists of two exons, were identified to be specifically expressed in monocyte-derived macrophages (MoDMs) in acute craniocerebral injury. Inhibition of Maclpil markedly decreased the expression of proinflammatory genes while upregulating the expression of anti-inflammatory genes. The lncRNA Maclpil plays a major role in the phenotypic transformation of macrophages by downregulating the expression of LCP1 [53].

### 2.9. MALAT1

Metastasis-associated lung adenocarcinoma transcript 1 (MALAT1), also referred to as NEAT2, a nuclear-retained long noncoding RNA with a length of 6983 nucleotides [54]. Both its high conservation in mammals and its abundant expression in many types of cells emphasize the importance of its function. MALAT1 expression was found to be markedly upregulated by LPS stimulation in PMA-induced THP-1 and RAW264.7 cells. MALAT1 knockout weakened the LPS-induced activation of M1 macrophages, revealing that MALAT1 plays a negative regulatory role in the NF-κB signaling pathway by binding the NF-κB subunit p65/50. MALAT1 could change the expression of inflammatory cytokines, including TNF-α and IL-6, and participate in the regulation of innate immunity and the inflammatory response [55]. Additionally, MALAT1 expression was significantly decreased after IL-4 treatment. Knocking out MALAT1 enhanced the differentiation of M2 macrophages [56].

### 2.10. MIR155HG

The lncRNA MIR155 host gene *(MIR155HG*) is located on chromosome 21q21 and is composed of three exons with a size of 12,079 nucleotides. *MIR155HG* expression significantly increased M1 macrophage polarization in chronic obstructive pulmonary disease induced by GM-CSF. Overexpression of *MIR155HG* upregulated the expression levels of M1 macrophage markers iNOS and CD86 and downregulated the expression of M2 macrophage markers ARG1 and CD206. *MIR155HG* could induce the expression of TNF-α, IL-1β and IL-12 and inhibit the secretion of IL-10 [57]. Thompson et al. found that the NF-κB p50/p65 heterodimer binds approximately 178 nt upstream of the *MIR155HG* promoter and induces its transcription, which suggests that *MIR155HG* is a key target gene of NF-κB [58].

### 2.11. PACER

The lncRNA P50-associated COX-2 extragenic RNA (PACER) is located upstream of the *Cox-2* gene, which is 2543 bp in length and contains a conserved recognition site for the CCCTC-binding factor CTCF. The expression of PACER and Cox-2 requires the induction of the CTCF/cohesin complex to control transcription initiation. When acting as a “decoy lncRNA” in U937 mononuclear macrophages stimulated by LPS, PACER mainly interacts with the NF-κB inhibitory subunit p50 and contributes to the restriction of protein binding to the Cox-2 promoter. PACER could regulate Cox-2 mRNA transcription and promote binding to the NF-κB p65/p50 dimer with activating ability. Krawczyk et al. demonstrated that PACER attracts a class of enzymes that are required for transcription activation. Additionally, knockout of PACER significantly affected the acetylation of histone H3 and H4 in LPS-induced macrophages. Additionally, PACER promotes P300 binding, chromatin opening, RNA polymerase II recruitment and finally transcriptional activation in macrophages [59].

### 2.12. PTPRE-AS1

Protein tyrosine phosphatase receptor type E (*PTPRE*) is a protein-coding gene. The lncRNA antisense to *PTPRE* (AS-PTPRE) has two variants. Variant 2 is referred to as PTPRE-AS1, which is 1494 nucleotides in length. Whole-genome sequencing revealed that PTPRE-AS1 could inhibit the expression of genes related to M2 macrophages, such as IL-10, Arg-1, and CD206, through epigenetic interference, and its expression was markedly upregulated in macrophages induced by IL-4. As a negative regulator of M2 macrophages, PTPRE-AS1 affects the proliferation and survival of macrophages by inhibiting the activation of the MAPK/ERK1/2 signaling pathway. Mechanistically, PTPRE-AS1 enriched in the nucleus directly interacts with WDR5 and H3K4 trimethylation, leading to transcriptional activation of the *PTPRE* gene. Sully et al. found that the level of TNF-α decreased and the level of IL-10 increased in *PTPRE*-gene-knockout BMDMs after LPS stimulation [60]. In dextran sulfate sodium (DSS)-induced murine colitis, PTPRE-AS1 deficiency has an anti-inflammatory effect and alleviates colitis symptoms [61].

### 2.13. RAPIA

The lncRNA Associated with the Progression and Intervention of Atherosclerosis (RAPIA), a lncRNA with a length of 10,252 nucleotides, is highly expressed in advanced atherosclerotic lesions and macrophages. SiRNA-mediated RAPIA silencing revealed that the activity of macrophage proliferation and colony formation decreased significantly when RAPIA was silenced, and flow cytometric analysis revealed an increase in the number of apoptotic cells. RAPIA located in the cytoplasm regulates the downstream target gene *ItgB1* (integrin β1) at the posttranscriptional level by binding to miR-183-5p. Inhibition of RAPIA can prevent advanced atherosclerosis and protect arteries [62].

### 2.14. ROCKI

LncRNA-Marcks or ROCKI (Regulator of Cytokines and Inflammation) consists 5377 nucleotides, is considered to be a major pro-inflammatory cytokine regulatory factor in human inflammation and is mainly distributed in the nucleus of THP-1 cells. Studies have shown that ROCKI expression can be highly induced by TLR4, TLR1/2, and TLR2/6 ligands. Overexpression of ROCKI induces a variety of cytokines and inflammatory genes, including ILR2, IL1R1, CXCL16, CXCL12, IL18B, MAPKAPK2, MAPK10, and MAPKAPK3. In contrast, knocking down ROCKI decreases the expression of IL-6 and IL-1α. ROCKI can form a ribonucleoprotein complex (RNP) with its binding protein APEX1 (apurinic/apyrimidinic endodeo xyribonuclease 1). The histone deacetylase HDAC1 could bind to the promoter of *Marcks*, resulting in a decrease in the expression of H3K27ac and *Marcks* and inhibition of the transcription of inflammatory factors [63]. Taken together, these findings indicate that as a negative regulator of the protein-coding gene *Marcks*, ROCKI plays a significant role in the innate immune response.

### 2.15. SNHG16

The lncRNA small nucleolar RNA host gene 16 (*SNHG16*) is located on human chromosome 17q25.1 and consists of 7573 nucleotides [64]. In THP-1 cells, overexpression of *SNHG16* significantly induced the expression of TNF-α, IL-1β, and IL-6 and promoted the proliferation of macrophages. An et al. reported that lncRNA-SNHG16 is a ceRNA that sponges miR-17-5p and promotes the expression of IKKβ, p-IκBα, and p-P65 in the NF-κB pathway [65]. Wang et al. found that *SNHG16* is mainly enriched in the cytoplasm in LPS-induced RAW264.7 cells. *SNHG16* is a negative ceRNA regulator that negatively regulates miR-15a/16 [66] and positively regulates inflammation induced by LPS.

## 3. Anti-Inflammatory Macrophage-Associated lncRNAs

### 3.1. ANCR

LncRNA-ANCR consists of 855 nucleotides. It plays key roles in preventing differentiation [67]. When ANCR is knocked out in macrophages, the expression of the M1 macrophage polarization markers IL-6 and IL-1β increases dramatically. ANCR can inhibit the expression of FoxO1 in THP-1 cells by promoting the ubiquitination and degradation of FoxO1 [68], which ultimately promotes the invasion and metastasis of gastric cancer cells [69].

### 3.2. LncRNA-AK085865

AK085865, a lncRNA with a length of 1,265 nucleotides, is located on mouse chromosome 6 and is transcribed from the second intron of the protein-coding gene *PPARγ*. Previously, Pei et al. analyzed the expression of lncRNAs in M1 and M2 BMDMs by microarray and found that lncRNA-AK085865 was highly expressed in asthmatic mice and was closely associated with M2 macrophages [70]. The latest research on the function of lncRNA-AK085865 in the occurrence and development of viral myocarditis (VM) induced by coxsackie virus B3 (CVB3) showed that the expression of lncRNA-AK085865 located in the nucleus in macrophages was significantly higher than that in monocytes. AK085865^−/−^mice promoted the polarization of M1 macrophages and susceptibility to VM [71]. A mechanistic study revealed that the 30–466 nt region of AK085865 specifically binds to interleukin enhancer-binding factor 2 (ILF2) and plays a negative regulatory role in the miR-192 pathway mediated by the ILF2-ILF3 complex, thereby promoting the polarization of M2 macrophages and reducing the symptoms of myocarditis [72].

### 3.3. LncRNA-CCL2

The lncRNA-CCL2 is a two-exon transcript spanning 32,792 bp of DNA [73]. Mouse lncRNA-CCL2 can be induced in activated macrophages to regulate innate and adaptive immunity [74]. In a mouse model of sepsis and in LPS-stimulated macrophages, the expression of lncRNA-CCL2 is increased dramatically, while that of Sir2 homolog (SIRT1) is decreased. After knocking down lncRNA-CCL2, the expression of proinflammatory genes was reduced [75]. Previous studies have shown that SIRT1, initially characterized as a histone deacetylase dependent on NAD^+^ coenzyme, is also a main target that is related to NF-κB. Activation of SIRT1 can reduce the level of NF-κB p65 acetylation. Jia et al. used siRNA to knockdown SIRT1 in LPS-stimulated macrophages and found that the expression of lncRNA-CCL2 and that of the proinflammatory cytokines IL-1β, IL6, and TNF-α increased. Downregulation of SIRT1 expression regulates immune function through lncRNA-CCL2 and histone modifying enzymes. Additionally, SIRT1 regulates the production of proinflammatory cytokines by inhibiting the expression of lncRNA-CCL2 [76].

### 3.4. Dnmt3aos

The lncRNA DNA methyltransferase 3A opposite strand (Dnmt3aos), a lncRNA with a length of 2952 nucleotides, is located on the antisense chain of the *DNMT3A* gene. Dnmt3aos functions as a chief regulator of *DNMT3A* expression. Li et al. reported that differentially expressed Dnmt3aos is closely associated with macrophage polarization in BMDMs stimulated by LPS + IFN-γ and IL-4. They found that the expression of nuclear Dnmt3aos in M2 (IL-4) macrophages was significantly upregulated compared with that in M1 (LPS + IFN-γ) macrophages, as measured by RT-qPCR. Dnmt3aos plays an indispensable role in promoting the phenotypic transformation of inflammatory macrophages to tissue-healing macrophages. Methylated DNA immunoprecipitation sequencing and mRNA expression profile analysis revealed that the Dnmt3aos-DNMT3A axis regulates the polarization of macrophages [77].

### 3.5. LincRNA-EPS

LincRNA-erythroid prosurvival (EPS), also known as Ttc39aos1, is 2347 nucleotides in length. It localizes to the regulatory regions of immune response genes (*IRGs*). Many studies have shown that lincRNA-EPS expression is decreased in mouse BMDMs treated with LPS. Inflammatory symptoms were prominent in lincRNA-EPS knockout mice. A mechanistic study revealed that binding of the CANACA motif at the 3′ end to hnRNPL is one function of lincRNA-EPS located in the nucleus of resting macrophages. Through binding to this motif, LincRNA-EPS enhanced the binding of inflammatory chemokines (such as Cxcl10, Cxcl9, and IL27) to the transcriptional initiation sites of IFN-stimulating genes. Further research revealed that lincRNA-EPS is an indispensable regulator of the expression of IRGs and the host’s defense against pathogens [78]. LincRNA-EPS maintains the suppressed chromatin state by controlling the localization of nucleosomes, thereby suppressing the transcription of IRGs. These results collectively indicate that lincRNA-EPS functions as an immunoregulatory lncRNA that can inhibit the activation of immune regulatory genes [79].

### 3.6. Kcnq1ot1

Kcnq1ot1 is an antisense lncRNA located on chromosome 11 at p15.5 [80]. An RNA polymerase II (RNAPII)-encoded 93,092 nt-long transcript. In primary mouse BMDMs and RAW264.7 cells, after polymethyl methacrylate (PMMA) stimulation in vivo and in vitro, the expression of Kcnq1ot1 was decreased. Overexpression of Kcnq1ot1 could reduce TNF-α and iNOS expression while increasing IL-10 and Arg1 expression, inducing M2 macrophage polarization [81]. In addition, the expression of miR-21a-5p, which can target IL-10, is upregulated and acts as a decoy molecule of Kcnq1ot1 in RAW264.7 cells. The lncRNA Kcnq1ot1 was found to interact with chromatin and the H3K9 and H3K27 specific histone methyltransferase G9a and PRC2 complex members (Ezh2 and Suz12) in a lineage-specific manner and recruit these proteins to the promoters of flanking genes to establish repressive chromatin marks [82]. A targeted deletion encompassing an 890 bp silencing domain of the Kcnq1ot1 promoter led to selective relaxation of the imprinting of ubiquitously imprinted genes on the paternal allele. It has also been proposed that the 890 bp region regulates the imprinting of ubiquitously imprinted genes by maintaining DNA methylation of somatic differentially methylated regions (DMRs) through interacting with Dnmt1 [83].

### 3.7. Lethe

Lethe was the first confirmed functional pseudogene lncRNA located on mouse chromosome 4 and consists of 501 nucleotides [84]. Zgheib et al. found that high glucose conditions decreased Lethe expression while significantly increasing NADP oxidase 2 (NOX2) and reactive oxygen species (ROS) expression in RAW264.7 cells. Lethe overexpression suppressed the production of ROS in diabetic primary macrophages mediated by NOX2. Mechanistically, Lethe regulates the interaction of the p65-NF-κB complex with the NOX2 promoter in the nucleus and blocks the binding of P65 to DNA and the activation of downstream genes, which ultimately has an anti-inflammatory effect [84].

### 3.8. Mirt2

Myocardial infraction-associated transcript 2 (Mirt2) is located on mouse chromosome 15 and is 3379 nucleotides in length. Du et al. found that Mirt2 localized to the cytoplasm of mouse peritoneal macrophages after stimulation with LPS. They found that Mirt2 was the most remarkably upregulated and highly expressed lncRNA after LPS treatment. In peritoneal macrophages, the activation of LPS-P38-Stat1 and the LPS-IFN-α/β-Stat1 pathway promote the transcription of Mirt2. TNF receptor-related factor 6 (TRAF6) governs the regulation of Mirt2-mediated inflammation. Mirt2 could bind and attenuate the ubiquitination of TRAF6 at lysine 63 (K63), thus inhibiting the activation of NF-κB and mitogen-activated protein kinase pathways and limiting the production of proinflammatory cytokines (including TNF, IL-1β, IL-6, and IL-12). In conclusion, as a negative regulator, Mirt2 can alleviate the inflammatory response induced by LPS by inhibiting TRAF6 oligomerization and autologous ubiquitination [85].

### 3.9. MIST

Macrophage Inflammation Suppressing Transcript (*MIST*) consists of 577,802 nucleotides. Previous studies on obesity and diabetes have confirmed that NF-κB is activated in adipose tissue and macrophages and that the expression of the proinflammatory cytokines TNF-α and IL-6 is increased. In a recent study using obese mice, Kenneth et al. found that the expression of some lncRNAs was downregulated in macrophages, one of which was a macrophage-specific intergenic lncRNA referred to as lncRNA-MIST. *MIST* is located 5.6 kb downstream of the protein-coding gene fatty acid binding protein 5 (Fabp5). Proinflammatory cytokine expression was upregulated after *MIST* knockout, while anti-inflammatory cytokine (Egr2, CD83, and PPARD) expression and Fabp5 expression were inhibited. Mechanistically, the MIST-PARP1 (poly ADP-ribose polymerase-1) interaction could block the recruitment of PARP1 to the promoters of inflammatory genes and then reduce the expression of those genes. Therefore, *MIST* is a negative regulator of inflammation. In addition, Sun et al. reported that *MIST* functions as an epistatic regulator of Fabp5 in cis and affects gene expression in *trans* [86].

### 3.10. LncRNA-Mm2pr

The lncRNA macrophage M2 polarization regulator (Mm2pr) is located between the *Hmgb1* and *Usp11* genes on chromosome 5. It consists of three exons with 1715 nucleotide transcripts. In mouse BMDMs and RAW264.7 cells, the expression of lncRNA-Mm2pr was remarkably upregulated after IL-13 and IL-4 stimulation. In LPS-induced polarized M1 macrophages, the expression of lncRNA-Mm2pr was significantly downregulated. Moreover, after silencing lncRNA-Mm2pr in BMDMs and RAW264.7 cells, the expression of CD206 decreased significantly after stimulation by IL-4 and IL-13 [5]. LncRNA-Mm2pr plays an important role in the polarization of M2 macrophages with immunomodulatory activity. Moreover, lncRNA-Mm2pr promotes the occurrence and development of tumors through the phosphorylation of STAT6 [24].

### 3.11. NEAT1

Nuclear Enriched Abundant Transcript 1 (NEAT1) is a kind of “architectural RNA” located on chromosome 11 with 3418 nucleotides [87]. NEAT1 functions as the core or scaffold of nuclear bodies (NBs) and is essential for the structure of paraspeckles [88]. Gao et al. found that miR-214, which is a target molecule of NEAT1, could directly interact with B7-H3 and promote the polarization of M2 tumor-associated macrophages in multiple myeloma. NEAT1 promoted the occurrence and development of multiple myeloma through the JAK2/STAT3 signaling pathway [89]. NEAT1 induces the expression of antiviral genes, such as IL-8, by TLR3 and plays a regulatory role in the innate immune response via inflammatory cytokines [90]. In addition, NEAT1 also participates in the differentiation of mononuclear macrophages [91].

## 4. Conclusions

With the rapid progress of high-throughput sequencing and bioinformatics, various crucial functional roles for noncoding RNAs in human development and diseases have been gradually revealed. LncRNAs participate in the regulation of almost every stage of gene expression in different diseases, functioning as signal molecules, bait molecules, guide molecules, and skeleton molecules [92]. Numerous studies have revealed that lncRNAs are central factors in genomic imprinting, chromatin modification, transcriptional regulation, gene posttranscriptional regulation, splicing, and modification [2,93,94,95,96,97]. The roles of lncRNAs in coordinating diverse aspects of the macrophage response are summarized in Table 1. LncRNAs specifically regulate macrophage polarization through diverse regulatory mechanisms (summarized in Figure 1) and mediate the occurrence and development of a variety of diseases. For instance, lncRNA-COX2 is significantly associated with increased susceptibility to sepsis. Dnmt3aos is involved in type 2 diabetes [98]. GAS5 is not only related to diabetes but may also be an effective target for atherosclerosis treatment [99]. Lethe expression is closely related to wound healing in patients with diabetes [84]. The expression of H19 and RAPIA has been found to be increased in patients with atherosclerosis [62,100]. LncRNAs NEAT1 and PTPRE-AS1 are involved in the occurrence and development of inflammatory bowel disease [61,87]. In addition, the lncRNAs Maclpil and MIR155HG are involved in acute brain injury [53,101] and chronic obstructive pulmonary disease [101], respectively. The genetic variation in lncRNA-ROCKI is linked to reduced risk for some human inflammatory and infectious diseases, including inflammatory bowel disease and tuberculosis [102], emphasizing the prominence of cis-acting lncRNAs in TLR signal transduction, innate immunity and pathophysiological inflammation. By intervening in the secretion of macrophage-related cytokines or the expression of inflammatory genes, lncRNAs alter the components of the inflammatory microenvironment and regulate the polarization of macrophages. Nevertheless, the regulatory mechanism of lncRNAs in M2 macrophage activation remains to be explained in detail. Additionally, lncRNAs have been implicated in the occurrence and development of several diseases and have been used for the diagnosis and treatment of these diseases, including nervous system diseases, cardiovascular diseases, and immune metabolic diseases [103,104].

Innate immunity provides a robust first line of defense against pathogenic microorganisms [105]. The TLR/NF-κB signaling pathway plays a key role in the regulation of innate immunity and is tightly regulated [106]. Some lncRNA expression levels change in conjunction with the activation of NF-κB signaling, which is a key step in the modulation of the complex NF-κB signaling pathway. Some lncRNAs, including PACER, Lethe, and MALAT1, directly interact with NF-κB to regulate target gene transcription. Furthermore, another group of lncRNAs, such as HOTAIR, CARLR, and MIR155HG (summarized in Figure 2), indirectly regulate NF-κB signaling activity by interacting with upstream components. On the other hand, for the proinflammatory macrophage-targeting lncRNAs, both PACER and MALAT1 are involved in the interaction with the p65-p50 complex to prevent NF-κB binding to promoter regions of NF-κB-responsive genes [55,59]. However, the other lncRNAs mentioned above (CARLR, HOTAIR, MIR155hg, and SNHG16) regulate NF-κB activity by interfering with its upstream components or related signaling molecules. Given the involvement of lncRNAs in NF-κB regulation, there is evidence of an association between lncRNAs (e.g., CARLR, HOTAIR, MIR155hg, and SNHG16) and human diseases relevant to NF-κB dysfunction [28,45,59,66]. Interestingly, lncRNAs are usually expressed on a limited subset of cells. This unique specificity is another advantage of further exploring lncRNAs as therapeutic targets of the NF-κB pathway [107].

Consequently, considering that lncRNAs participate in the regulatory network of NF-κB, RNA interference- or antisense RNA technique-mediated silencing of inflammatory lncRNAs associated with promoting the NF-κB signaling pathway may provide new insights for the diagnosis and treatment of diseases. Novel targeting of these lncRNAs could substantially improve the treatment of inflammatory diseases. In addition, CRISPR/Cas9 (clustered regulatory interspaced short palindromic repeats/CRISPR-associated protein 9) [108], a powerful rapid genome editing tool, could also be applied for the study of immunologic diseases. For example, lincRNA-COX2 and lincRNA-AK170409, which are associated with NF-κB signal transduction and were identified by the CRISPR/Cas9-mediated deletion of lncRNA sites, are expected to be used for the treatment of inflammatory diseases and for further elucidation of related mechanisms [109].

LncRNA functions as biomarkers in the diagnosis of immune diseases are still under investigation. Understanding the relationship between lncRNAs and different clinical stages of diseases could improve current clinical diagnostic and treatment strategies. Therefore, it is not surprising that the study of lncRNAs has quickly become a cutting-edge field in molecular biology. However, due to the limitations of technology and research methods as well as the fact that lncRNAs are not highly conserved, our current knowledge of lncRNAs is just the tip of the iceberg. There is an urgent need to explore the countless treasures that remain to be uncovered in the lncRNA field.

## Figures and Tables

**Figure 1 cells-11-00005-f001:**
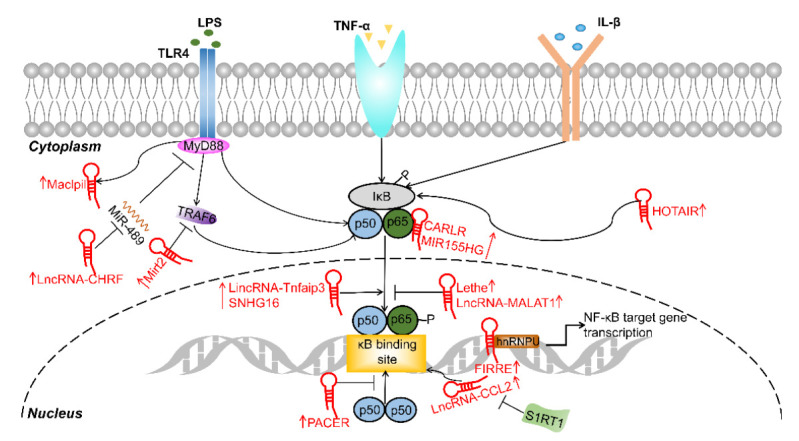
LncRNAs regulate the NF-κB signaling pathway. LncRNA-CHRF, Mirt2, Lethe, MALAT1, and PACER suppress the NF-κB signaling pathway. LincRNA-Tnfaip3, SNHG16, CARLR, MIR155HG, HOTAIR, and lncRNA-CCL2 active the NF-κB signaling pathway.

**Figure 2 cells-11-00005-f002:**
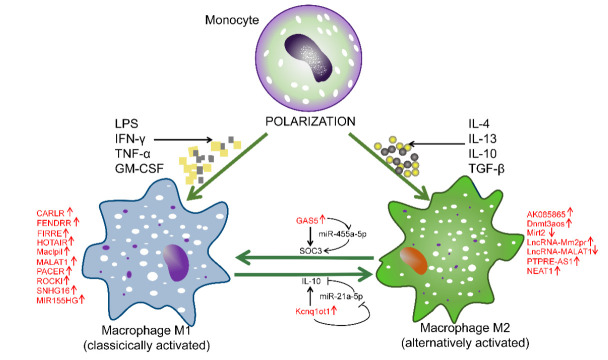
LncRNAs regulate the polarization of M1 and M2 macrophages. LncRNA-CARLR, CHRF, FENDRR, FIRRE, HOTAIR, H19, Maclpil, MALAT1, MIR155HG, PACER, ROCKI, GAS, and SNHG16 induce macrophage polarization into the M1 phenotype. LncRNA-AK085865, Dnmt3aos, PTPRE-AS1, Mm2pr, KCNQ1OT1, and NEAT1 induce macrophage polarization into the M2 phenotype.

**Table 1 cells-11-00005-t001:** Summary of lncRNAs in macrophages.

LncRNA Names	Species	Classification	Localization	Role and Function in Macrophages	Possible Mechanism	References
CARLR	Mouse	intergenic	Cytoplasm	Increased after activation of NF-κB in macrophage	Binds to p65 and lets it Released from IκBα	[28]
CHRF	Mouse	intergenic	Cytoplasm	Acts as the endogenous “sponge” of miR-489	Changes the expression of MyD88	[29]
FENDRR	Human Mouse	intergenic	Nucleus	Overexpression of FENDRR increases mRNA expression of M1 markers induced by IFN-γ in primary mouse BMDMs	Enhances IFN-γ induced M1 macrophage polarization via the STAT1 pathway.	[33]
FIRRE	Human	intergenic	Nucleus	Induces inflammatory gene expression in macrophages	Inhibits chromatin state through interaction with hnRNPU	[34,35]
GAS5	Human Mouse	antisense	Nucleus	Induces polarization of M1 macrophages and aggravates inflammation	Acts as the ceRNA of miR-455-5p and a sponge of miR-221	[42,43]
HOTAIR	Human	antisense	Cytoplasm	Upregulated after LPS stimulation in RAW264.7 cells and BMDM	Degrades expression of IκBα, activates NF-κB, and translocates the position of nucleus	[45]
H19	Human Mouse	intergenic	Nucleus	Upregulates the expression of IL-6, IL-1β, COX-2, CCL-5	Promotes the differentiation and activation of pro-inflammatory M1 macrophages	[52,53]
Maclpil	Mouse	-	-	Upregulates its expression under the stimulation of LPS in MoDM and MiDM	Polarizes pro-inflammatory macrophages	[54]
MALAT1	Human Mouse	intergenic	Nucleus	Upregulated after the action of LPS and in PMA-induced THP-1 cells and RAW264.7 cells	Binds NF-κB subunit p65/50	[56]
MIR155HG	Mouse	-	Nucleus	Upregulated in macrophages induced by GM-CSF	Induced polarization of M1 macrophages	[59]
PACER	Human	antisense	Nucleus	Upregulated following stimulation of LPS in U937 monocyte derived macrophage	Regulates chromatin acetylation	[60]
PTPRE-AS1	Human Mouse	antisense	Nucleus Cytoplasm	Upregulated in IL-4 induced macrophages and inhibit the expression of IL-10, Arg-1, and CD206	Inhibits the activation of MAPK/ERK1/2 signal pathway	[61]
RAPIA	Mouse	-	Cytoplasm	Promotes proliferation and reduces apoptosis of macrophages	Combines with miR-183-5p and regulates downstream ITGB1	[63]
ROCKI	Human		Nucleus	Induces a variety of cytokines and inflammatory genes	HDAC1 binds to the Marcks promoter, reducing the expression of H3K27ac and Marcks	[64]
SNHG16	Human	intergenic	Cytoplasm	Induces IL-6, TNF-α, and IL-1β expression in THP-1 cells	Acts as a ceRNA to absorb miR-17-5p to promote IKKβ, p-IκBα, and p-P65 in NF-κB pathway	[66]
ANCR	Human	intergenic	Cytoplasm	Anti-differentiation function	Downregulates the expression of FoxO1 and inhibits the polarization of M1 macrophages	[69]
LncRNA- AK085865	Mouse	-	Nucleus	Highly expressed in asthmatic mice and closely related to M2 macrophages	Regulates ILF2-ILF3 complex-mediated miRNA-192 biogenesis	[73]
LncRNA- CCL2	Mouse	-	Cytoplasm	Leads to downregulation of inflammatory cytokines in macrophages	Inhibits Sirt2 homolog 1 (SIRT1)	[76]
Dnmt3aos	Mouse	antisense	Nucleus	Upregulated in M (IL-4) macrophages	DNA methylation caused by the Dnmt3aos-DNMT3A axis	[78]
LincRNA- EPS	Mouse	intergenic	Nucleus	Downregulated under the stimulation of TLR2 in BMDMs	Binds to chromatin hnRNPL through the CANACA motif located at the 3′ end of EPS	[79]
Kcnq1ot1	Mouse	antisense	Cytoplasm	Downregulated under the stimulation of PMMA in primary mouse BMDMand RAW264.7 cells	Acts as a ceRNA to absorb miR-21a-5p to M2 macrophage polarization	[83]
Lethe	Mouse	-	Nucleus	Induced by pro-inflammatory cytokines (TNF-α, IL-1β) leads to upregulation ofits expression in BMDMs	Interacts with p65 to repress the activation of NF-κB	[85]
Mirt2	Mouse	antisense	Cytoplasm	Prevents the abnormal activation of inflammation in peritoneal macrophages	Associates with TRAF6 and weakens the ubiquitination of its connection with lys63, thus inhibiting the activation of NF-κB and MAPK pathways	[86]
MIST	Human Mouse	intergenic	Nucleus	Downregulated in obese mice induced by high-fat diet	Interacts with PARP1 to block recruitment of PARP1 on the promoter of inflammatory gene	[87]
LncRNA- Mm2pr	Mouse	-	Nucleus	Upregulated after IL-13 and IL-4 stimulation in BMDM and RAW264.7 cells	Through the phosphorylation of STAT6 to promotes the occurrence and development of tumor	[5,24]
NEAT1	Human Mouse	intergenic	Nucleus	Promotes its target molecule miR-214 interacts with B7-H3	Via JAK2/STAT3 signal pathway to promote the occurrence of multiple myeloma	[90]

Abbreviations: ARG1, Arginase 1; BMDMs, bone marrow-derived macrophages; CeRNA, competing endogenous RNAs; GM-CSF, granulocyte-macrophage colony stimulating factor; hnRNPU, heterogeneous nuclear ribonucleoprotein U; ITGB1, integrin β1; ILF2, interleukin enhancer-binding factor 2; LPS, lipopolysaccharide; Lys63, Lysine 63; MoDM, monocyte-derived macrophages; MiDM, microglial-derived macrophages; PMMA, polymethyl methacrylate; PMA, phorbol 12-myristate 13-acetate; TRAF6, TNF receptor-associated factor 6.

## Data Availability

Not applicable.

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
