# Peer review of "Long Noncoding RNAs Regulate the Inflammatory Responses of Macrophages"

_cells, 2021, doi:10.3390/cells11010005_

Round 1

Reviewer 1 Report

Overall, the manuscript reads well and provides a comprehensive review of the regulatory role of lncRNAs in macrophages.    

I would suggest the following minor amendments, which might improve the manuscript.  

1 - It would benefit the authors to add a reference column to Table 1—summary of lncRNAs in macrophages.  

2 - a short paragraph describing the interplay between long non-coding RNAs and epigenetic marks in regulating the Immune response and innate immunity (e.g. trained Immunity) would be helpful. 

Author Response

Thank you! 

Best Wishes,

Wei Shao

Reviewer 2 Report

  1. This review summarizes the role of the known Long Noncoding RNAs (lnsRNAs) in the inflammatory responses of macrophages. The authors put everything together in a very concise way.

  1. There are some inconsistencies in describing all these different lncRNAs like in some authors mentioned the size, but in some, this information is missing. It would be good if all had similar information while describing.

  1. For the proinflammatory macrophage-targeting IncRNAs, they listed 15 IncRNAs and stated their roles in proinflammatory responses individually. However, whether there is any crosstalk among those IncRNAs is not clear. For example, Carlr and HOTAIR are both involved in the NF-kB signaling pathway, but it is unclear whether these IncRNAs work independently or have any connections.

  1. Some minor errors, for example, in the abstract, “physiological a pathological” should be “physiological and pathological.”

Author Response

Thank you! 

Best Wishes,

Wei Shao
